# Mechanically Stimulated Solid-State Interaction of Platinum Tetrachloride with Sodium β-Diketonates

**DOI:** 10.3390/molecules28083496

**Published:** 2023-04-15

**Authors:** Victor D. Makhaev, Larisa A. Petrova

**Affiliations:** Federal Research Center of Problems of Chemical Physics and Medicinal Chemistry, Russian Academy of Sciences, Chernogolovka 142432, Russia

**Keywords:** platinum, β-diketonates, mechanochemistry, solid-state synthesis, green chemistry

## Abstract

A new mechanically stimulated solid-state reaction of PtCl_4_ with sodium β-diketonates has been discovered. Platinum (II) β-diketonates were obtained by grinding excess sodium trifluoroacetylacetonate Na(tfac) or hexafluoroacetylacetonate Na(hfac) in a vibration ball mill, followed by subsequent heating of the resulting mixture. The reactions occur under much milder conditions (at about 170 °C) compared to similar reactions of PtCl_2_ or K_2_PtCl_6_ (at about 240 °C). Excess diketonate salt plays the role of a reducing agent in the conversion of Pt (IV) salt to Pt (II) compounds. The effect of grinding on properties of the ground mixtures was studied by XRD, IR, and thermal analysis methods. The difference in the course of the interaction of PtCl_4_ with Na(hfac) or Na(tfac) indicates the dependence of the reaction on the ligand properties. The probable reaction mechanisms were discussed. This method of synthesis of platinum (II) β-diketonates makes it possible to substantially reduce the variety of reagents used, the number of reaction steps, the reaction time, the use of solvents, and waste generation compared to conventional solution-based methods.

## 1. Introduction

Due to their unique properties, platinum-group metals β-diketonates and their derivatives are increasingly used in the chemical industry, microelectronics, optics, hydrogen energy systems, medicine, and other fields [1,2,3,4,5,6,7]. Conventional methods for the synthesis of platinum metal β-diketonates are based on the interaction of β-diketone and a metal salt in aqueous or aqueous organic solutions [8,9,10]. As stated earlier by Chimitov et al. [11], there is currently no general method for the synthesis of platinum-group metal β-diketonates, such as M(β-diketonate)_2_ (M = Pd, Pt) and M(β-diketonate)_3_ (M = Ru, Rh, Ir, Os). In this regard, the possibilities of using alternative technologies for their preparation, e.g., microwave-assisted synthesis [11] or solid-state synthesis [12], are explored by researchers. Note that the use of various mechanochemical techniques (solvent-free/solvent-less processes, liquid-assisted grinding) in many cases makes it possible to accelerate reactions, increase the efficiency of processes, and eliminate or significantly reduce the use of solvents and waste generation. The mechanochemical approach is more environmentally friendly than conventional solution-based reactions [13,14,15,16,17]. In 2019, IUPAC included mechanochemistry in the list of top ten technologies for sustainable development [18,19]. A comprehensive review on the use of alternative technologies for the synthesis of various types of organometallic and coordination compounds has recently been published by Beillard et al. [20].

Recently, we developed a new general solid-state mechanochemical method for the synthesis of platinum-group metal β-diketonates by reacting metal chlorides (PdCl_2_, PtCl_2_, RuCl_3_^.^*n*H_2_O, RhCl_3_^.^*n*H_2_O, K_2_PtCl_6_) with β-diketone salts [12]. However, we were not satisfied with the high temperature (ca. 240 °C) required for the formation of Pt(β-diketonate)_2_ from ground mixtures with a simple PtCl_2_ or K_2_PtCl_6_ complex salts. Therefore, our goal was to find solid-state reactions for the synthesis of Pt(β-diketonate)_2_ under much milder conditions. To solve this problem, we studied the solid-state interaction of a simple PtCl_4_ salt with sodium trifluoroacetylacetonate Na(tfac) or hexafluoroacetylacetonate Na(hfac) using mechanical activation (ball milling) in the absence of a solvent, and compared the reactivity of PtCl_2_, K_2_PtCl_6_, and PtCl_4_ in the reactions.

## 2. Results

Grinding a PtCl_4_–4.5 Na(hfac) mixture in a vibration ball mill for 3–4 h produced a yellow powder, which turned brown when heated to ca. 150 °C. Strong reflections of the reactant Na(hfac), weak reflections due to PtCl_4_, and traces of the products NaCl and Pt(hfac)_2_, as well as possible intermediates (PtCl_2_) or by-products, were observed in the X-ray powder diffraction patterns of the mixture after grinding. With increased grinding time, the intensity of the NaCl reflections only slightly increased, while that for the reactant PtCl_4_ decreased (Figure 1).

Only slightly shifted absorption bands of the reactant Na(hfac) were observed in the IR spectra of the mixture after grinding (Figure 2, Table 1). The IR data corresponded to the results of XRD analysis, and indicated that the reaction passed through a series of intermediate stages and led to the formation of only traces of the final product under the grinding conditions used. The conclusion about the multistage nature of the process under study is consistent with modern ideas about the dynamics of mechanochemical processes, according to which solid-state mechanochemical reactions proceed in several stages, including grinding and physical mixing of reagents, their homogenization at the molecular or cluster level, and the conversion of the activated reactants into reaction products [24]. Earlier, we showed that the transformation of the activated mixture into the final products for a number of mechanically stimulated solid-state reactions occurs when the mixture is heated [25]. In this regard, we carried out a thermoanalytical study of the reaction mixtures. 

Thermal analysis of the PtCl_4_–4.5Na(hfac) reaction mixtures after 3–4 h grinding revealed several endothermic DTA peaks corresponding to weight losses in the TG curves. In the temperature range of ca. 50–150 °C, between 5 and 10% of weight was lost (it appears that Na(hfac) decomposition products arise from the Pt (IV)–Pt (II) reduction process). The intensity of the peaks increases markedly with an increase in the grinding time from 3 to 4 h. In our opinion, the data of instrumental investigations may indicate that the reaction proceeds through the stage of reduction of the ground PtCl_4_ to finely dispersed PtCl_2_ (d 6.48 Ǻ, 3.10 Ǻ, 1.90 Ǻ) (Figure 1).

It is known that solid PtCl_4_, when heated to 380 °C, decomposes with the release of chlorine. When grinding PtCl_4_, especially in the presence of a reducing agent, the decomposition temperature should decrease, which is what we observe in our reaction. To interact with one molecule of chlorine, one molecule of the reducing agent is sufficient. Fine dispersion of PtCl_2_ formed in this process leads to its high reactivity. To obtain Pt(β-diketonate)_2_, two moles of the diketone salt are needed. The ratio (ligand):(metal) = 4.5:1 was used. Here, 2 moles of the ligand went to the formation of the complex, 1 mole went to the reduction, and 1.5 moles were taken to ensure the completeness of the reaction. Weak endothermic peaks at about 70 °C and 110 °C, the intensity of which increased with increasing grinding time, indicated the formation of the decomposition products of the hexafluoroacetylacetonate anion. The formation of volatile products can slow down the course of this solid-state reaction due to the dissipation of mechanical energy, which is due to its consumption for the processes of evaporation of volatile products (see below). Therefore, to complete this solid-state reaction, it is necessary to heat the activated mixture.

The weak endothermic peak in the range of 150–220 °C with an extremum at ca. 180 °C corresponds to the main stage of the weight loss changing from 28% after 3 h to 44% after 4 h grinding. Visual observation shows that sublimation of the target product Pt(hfac)_2_ occurs in this temperature range. A weak sharp peak at ca. 230 °C corresponding to a weight loss of ~10% is presumably due to thermal degradation of Pt(hfac)_2_ and/or Na(hfac) (Figure 3a,b).

To avoid thermal decomposition and contamination with by-products formed upon the decomposition of Na(hfac) during Pt (IV)–Pt (II) reduction, the product Pt(hfac)_2_ was isolated by vacuum sublimation of the ground mixtures at 140–180 °C. The PtCl_4_–Pt(hfac)_2_ conversion after grinding for 4 h reached 71% (vide infra, Experimental). The IR and XRD data for the product are consistent with published data for Pt(hfac)_2_ [8,12]. The residue after the isolation of Pt(hfac)_2_ showed intense NaCl reflections and weak Na(hfac) reflections (Figure 1e).

Comparison of the PtCl_4_–4.5Na(hfac) reaction to previously studied Na(hfac) reactions with PtCl_2_ or K_2_PtCl_6_ (Figure 3c,d) [12] demonstrated that the reaction with PtCl_4_ proceeded under much milder conditions (at a temperature ~50–60 °C or more lower than with PtCl_2_ or K_2_PtCl_6_) and results in the formation of Pt(hfac)_2_ with a higher conversion degree.

Thus, the processes occurring during this reaction can be represented by Figure 1:

The solid-state reaction of PtCl_4_ with Na(tfac) differs somewhat from the reaction described above. Grinding PtCl_4_–4.5Na(tfac) for 3–4 h leads to the formation of a brown PtCl_4_–4.5 Na(tfac) mixture.

The X-ray diffraction pattern of the ground mixture consists mainly of NaCl reflections, an amorphous phase at 2θ~20°, and traces of the reactants Na(tfac) and PtCl_4_ (Figure 4c). The presence of strong NaCl reflections (Figure 4) may indicate the formation of substitution products in the ground mixtures. The formation of substitution products also manifested in the IR spectra by a slight shift in the absorption bands of the Na(tfac) C=O group (Table 2), and indicated a difference in the mechanisms of PtCl_4_ interaction with Na(hfac) and Na(tfac). Mainly, strong reflections of the product NaCl and weak reflections of the reactant Na(tfac) were observed in the residue after Pt(tfac)_2_ sublimation.

The IR spectra mainly display slightly shifted absorption bands of Na(tfac) and some bands close to those of Pt(tfac)_2_ (Figure 5, Table 2).

The DTA curve of the reaction mixture after grinding for 4 h shows weak endothermic peaks at 98 and 110 °C corresponding to a weight loss of ~4% in the TG curve (Figure 6). In the range of 110–180 °C, an intense exothermic peak was observed with a maximum at ~170 °C, corresponding to the main stage of weight loss. Visual observation shows that the vacuum sublimation of the target product, light yellow Pt(tfac)_2_, occurs in the range of 140–180 °C. The degree of conversion of PtCl_4_ to Pt(tfac)_2_ was ~69% (vide infra, Experimental).

Comparison of the results from instrumental research methods suggests that, unlike reaction in the PtCl_4_–4.5Na(hfac) mixture (Figure 1, formation of PtCl_4_ reduction products), this reaction proceeds via a sequence of addition–elimination steps with the possible formation of intermediates such as Na[PtCl_4_(tfac)]^+^, Pt(tfac)_2_Cl_2_ or others, and the production of the desired Pt(tfac)_2_. Taking into account the polymeric structure of PtCl_4_, the formation of various other intermediates can be assumed [26].

Thus, the processes occurring during this reaction can be represented by Figure 2:

A difference in the course of PtCl_4_ reactions with Na(hfac) (Figure 1) and Na(tfac) (Figure 2) indicates a significant dependence of the reaction course on the structural, electronic, and steric properties of organic ligands. Na(hfac) easily reduces PtCl_4_ with the formation of highly volatile products and PtCl_2_, while the reaction with Na(tfac) proceeds via addition–elimination steps. It can be expected that PtCl_4_ reactions with salts of less acidic β-diketones will also proceed similarly to the reaction with Na(tfac).

Thermal analysis data show that the main stage of weight loss (sublimation of target products) corresponds to the weak endothermic effect, with an extremum at ca. 178 °C for the reaction of PtCl_4_ with Na(hfac) (Figure 3) and the exotherm with a maximum at ca. 171 °C for the reaction with Na(tfac) (Figure 6); this is in contrast to the analogous reactions with PtCl_2_ or K_2_PtCl_6_ accompanied by exothermic effects with maxima at ca. 240 °C [12]. These data indicate a decrease in the formation temperature of Pt(II) diketonates by about 60 °C when using PtCl_4_ instead of PtCl_2_ or K_2_PtCl_6_.

The degree of conversion of the initial platinum salt in the considered solid-state reactions is comparable to that for conventional solution-based methods [8]. However, the solid-state synthesis approach makes it possible to achieve the following: significantly reduce the variety of reagents used, the number of reaction steps, and the reaction time; reduce or eliminate the use of solvents; and significantly reduce waste generation. Further research is necessary to elucidate the mechanism and optimize the conditions of the reactions under consideration.

Previously, in the course of a systematic study, we studied the feasibility and some features of the solid-state mechanochemical synthesis of various classes of coordination and organometallic compounds, such as cyclopentadienyl complexes of *d*- [27] and *f*- elements [28], Fe, Co, and Ni metallocenes [29], bis-dicarbollyl complexes of Fe, Co, and Cr [30], β-diketonates [25,31], carboxylates [32], dialkyldithiocarbamates [33], and other complexes of a number of metals.

The data obtained by us and by other researchers make it possible to reveal some typical features that make a difference between solid-state mechanochemical reactions and other mechanochemical technologies and conventional solution-based reactions. These features are associated primarily with the impact of mechanical energy on the crystalline reactants (grinding), which results in their physical mixing, fragmentation, and homogenization, an increase in the contact area between them, the formation of various defects in the crystal structure, and an increase in temperature of the reaction mixture. All these factors increase the reactivity of the reactants.

The accumulation of energy by mixtures during their grinding can be most clearly detected using thermal analysis. During grinding, a new exothermic effect appears in the DTA curves of the reaction mixtures, which is absent in the DTA curves of the reactants and products. With an increase in grinding time, its temperature decreases due to the formation of smaller particles of reactants in the mixture. This exotherm corresponds to the interaction of the reactants with the formation of final products. When the reaction temperature approaches the ambient temperature, the reactants are instantly converted into products in the thermal explosion mode [34].

Milled mixtures with too high temperatures of the exothermic effect (above 200 °C) do not reach the stage of transformation into final products even with prolonged grinding under experimental conditions. Instrumental research methods indicate the presence of only ground reactants in such mixtures (broadened absorption bands in the IR spectra and broadened reflections in powder X-ray diffraction patterns). Evidently, the power of the used vibration mill is not enough to heat the reactor to the temperature of the reaction initiation. These mixtures can be converted into products by further heating after grinding [12].

The yield of products is very sensitive to the quality of the reactants and decreases drastically in the presence of trace water or other volatile compounds because the presence or formation of liquid or viscous compounds prevents the transfer of mechanical energy to crystalline reactants due to energy dissipation. In this case, other regularities are observed [35].

In some cases, mixtures that have accumulated mechanical energy may remain reactive for some time, and they can be called activated mixtures. This allows them to be used to conduct or study reactions of the synthesis in the mode of self-propagation. The advantage of such systems is a low temperature of the process (less than 300 °C), and the possibility of visual control of the process due to a change in the color of the reaction mixture during the reaction. The reaction in the Na(acetylacetonate)–CrCl_3_ system is one of the first examples of the synthesis of complex compounds in the self-propagation mode. Reactions of this type demonstrate the relationship between the phenomena of mechanical activation, thermal explosion, and reactions in the self-propagation mode [25,34,36].

Most of the systems studied are characterized by the presence of both reaction products and a new exothermic effect in the DTA curves of milled mixtures. In these systems, both final or intermediate products and activated mixtures of reactants are formed as a result of grinding. On the thermograms of these mixtures, an exotherm is observed, the temperature of which decreases with an increase in the grinding time, but does not reach the inner temperature (temperature of the reactor). The superposition of the exothermic effects of the reaction and the endothermic effects of melting or sublimation of the products can lead to a complex shape of the DTA curves. In the X-ray powder diffraction patterns of the milled mixtures, the intensity of the reflections of the reactants decreases and that of the reaction products increases. The IR spectra indicate a decrease in the intensity of the absorption bands of reactants and an increase in the absorption bands of products. There is a gradual decrease in the amount of reactants, and an increase in the amount of reaction products in the ground mixtures. The reactions slow down due to dilution of the reaction mixture by the reaction products. In this case, heating of ground mixtures can significantly reduce time of synthesis, increase the product yield, and prevent mechanochemical decomposition of the target products [33].

The data obtained suggest that activating devices of a vibration-mill type are efficient enough for creating optimal conditions for mechanochemical synthesis of coordination and organometallic compounds. More powerful devices, such as planetary mills, are hardly suitable for the purpose since they can cause decomposition of the product and a decrease in the yield. Similar patterns in relation to the use of apparatuses of various types were observed in the synthesis of boron hydride derivatives [37].

A comparison of the three considered types of reactions indicates that the difference between them is of no fundamental importance and is caused only by the influence of external factors, namely, the temperature of the process. It can be assumed that when milling at low temperatures, the transformation of the activated mixture into products will be hindered, while at elevated temperatures most of the reactions will proceed in a purely mechanochemical mode.

The advantages of solid-state methods also lie in the fact that they make it possible to solve a number of problems that are difficult or impossible to solve by conventional methods, such as the preparation of non-solvated or hydrolysis-sensitive products [38], as well as products of complete replacement of chloride ligands in platinum compounds under mild conditions (this work) compared to reactions in solution or liquid-assisted grinding reactions [39].

Thus, the study of the effect of grinding on mixtures of solid metal chlorides with anionic derivatives of organic compounds made it possible to expand the existing ideas on the reactivity of solids, develop solid-state methods for the synthesis of certain classes of compounds, and discover a number of phenomena previously unknown in the chemistry of coordination and organometallic compounds.

## 3. Experimental

### 3.1. Materials and Methods

Sodium trifluoroacetylacetonate Na(tfac) and hexafluoroacetylacetonate Na(hfac) were obtained as described previously [31]. Freshly prepared β-diketonates were used in the reactions. Platinum (IV) chloride (Sigma-Aldrich; 96% purity) was used without further purification. Preparation of mixtures of starting materials, samples for instrumental studies, loading and unloading of a ball mill grinding jar, as well as other operations with air-sensitive substances were carried out in a nitrogen atmosphere in a drybox.

Grinding the reaction mixtures was carried out in a stainless-steel grinding jar with a volume of 85 cm^3^ using a custom-made vibration ball mill [12]. Weighed amounts of the reactants (~1 g) and 20 steel balls 12.3 mm in diameter (~150 g) as milling bodies were loaded into the jar. The jar was sealed, mounted on the vibration ball mill, and subjected to vibration (amplitude 11 mm, frequency 12 Hz; energy intensity of ~0.5 W/g) for a preset time. Then, the jar was opened and the reaction mixture was separated and used for instrumental studies and isolation of target products. The properties of the reactants, reaction mixtures, and products were studied by X-ray phase analysis, IR spectroscopy, and thermal analysis. The products were identified according to the data of chemical analysis and instrumental research methods. X-ray diffraction powder patterns of the samples were recorded on an ADN-2-01 diffractometer (CuKα radiation, Ni filter). IR spectra of the reactants, reaction mixtures, and products were recorded on a Bruker Vertex 70 v Fourier-transform infrared spectrometer. Thermal studies were performed on an STA-449 F5 Jupiter thermal analyzer (Netzsch, Selb, Germany) in the temperature range of 20–250 °C at a heating rate of 10 °C/min with a sample weight of ~10 mg.

### 3.2. Synthesis

Pt(hfac)_2_. A mixture of 0.2894 g (0.859 mmol) PtCl_4_ and 0.9003 g (3.906 mmol) Na(hfac) was ground for 4 h. The resulting yellow powder was studied by IR spectroscopy, XRD, and thermal analysis; part of the powder was used to isolate the product. From 0.7107 g of the activated mixture, 0.2226 g of Pt(hfac)_2_ was obtained as an orange crystalline substance by vacuum sublimation at 140–180 °C, ~10^−1^ Torr. PtCl_4_–Pt(hfac)_2_ conversion, 71%. M. p. 137–138 °C. Lit. data: 138 °C [40]. Found, %: C 19.67; H 0.35; Pt 32.1. Calculated for Pt(C_5_H_1_F_6_O_2_)_2_, %: C 19.69; H 0.33; Pt 32.13.

Pt(tfac)_2_. A mixture of 0.1858 g (0.5515 mmol) PtCl_4_ and 0.4775 g (2.713 mmol) Na(tfac) was subjected to grinding for 4 h. The resulting brown powder was studied by IR spectroscopy, XRD, and thermal analysis; part of the powder was used to isolate the product. From 0.2042 g of the reaction mixture, 0.0587 g of a light-yellow substance was obtained by vacuum sublimation at 150–190 °C. The IR and XRD data of the product correspond to a mixture of *cis*- and *trans*-Pt(tfac)_2_ isomers [8,40]. PtCl_4_–Pt(tfac)_2_ conversion, 69%. M. p. 172–173 °C. Lit. data: 170 °C [40]. Found, %: C 23.69; H 1.69; Pt 38.55. Calculated for Pt(C_5_H_4_F_3_O_2_)_2_, %: C 23.96; H 1.61; Pt 38.92.

## 4. Conclusions

In the course of a systematic study of the solid-state mechanochemical synthesis of coordination compounds, a new mechanically stimulated solid-state reaction of PtCl_4_ with excess Na(tfac) or Na(hfac) was discovered. The reaction, solid-state grinding PtCl_4_ with excess Na(tfac) or Na(hfac) in a vibration ball mill with subsequent heating, made it possible to obtain the corresponding Pt(II) β-diketonates in yields comparable to those of conventional solution-based methods and proceeded under much milder conditions compared to similar PtCl_2_ or K_2_PtCl_6_ reactions. Excess diketonate salt played the role of a reducing agent in the conversion of Pt (IV) salt to Pt (II) compounds. The effect of grinding on the properties of ground mixtures was studied by XRD, IR, and thermal analysis methods. The difference in the course of the reaction of PtCl_4_ with Na(hfac) or Na(tfac) indicated a dependence of the reaction on the ligand properties. Probable reaction mechanisms were discussed. In the case of the reactions under consideration, the solid-state synthesis approach makes it possible to eliminate the use of solvents, significantly reduce the variety of reagents used, the number of stages, the reaction time, and waste generation compared to conventional solution-based methods. An analysis of our data in addition to the literature made it possible to reveal some features of the mechanical activation of “metal chloride-anionic derivative of an organic compound” solid mixtures.

## Data Availability

The data presented in this study are available on request from the corresponding author.

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
