# Peer review of "Mechanically Stimulated Solid-State Interaction of Platinum Tetrachloride with Sodium β-Diketonates"

_molecules, 2023, doi:10.3390/molecules28083496_

Round 1
Reviewer 1 Report
Please add the synthetic scheme in the text.
Author Response
We added the schemes, and the probable reaction mechanisms were discussed in the revised version.
Reviewer 2 Report
This manuscript describes the mechanochemical preparation of Pt(II)-beta-diketonate complexes.
Here are several points that need to be clarified:
1) How the authors determined that the molar ratio 4.5 (ligand): 1 (metal) was adequate for the reaction to take place. Did you explore other stoichiometries to find this relationship the right one?
2) The slightly labile nature of the Pt-Cl bond has been mentioned in the literature, which represents problems from the point of view of mechanochemical reactions.1,2 In this sense, precursors with more labile ligands [PtCl2(PhCN)2] have been used. Haven't the authors used these to better proceed with the reaction? 3) Regarding point 2, have the authors explored LAG (liquid-assisted grinding) reactions? The addition of small amounts of solvent (in catalytic amounts) favors the reactions to proceed more adequately.3 4) How did you determine the reaction times (3 or 4 h)? Did the authors follow up on previous reaction times to determine these reaction times? 5) In the IR results, it would be appropriate if they mentioned the frequency of C=O of the diketonates as free ligands and once coordinated to corroborate the coordination of the ligand. 6) Regarding the characteristics of the ball-milling device, its brand was not specified. The authors are requested to clarify these observations. Therefore, by doing this, the publication of this manuscript can be recommended.(1) Adams, C. J.; Colquhoun, H. M.; Crawford, P. C.; Lusi, M.; Orpen, A. G. Solid-State Interconversions of Coordination Networks and Hydrogen-Bonded Salts. Angew. Chemie - Int. Ed. 2007, 46 (7), 1124–1128. https://doi.org/10.1002/anie.200603593.
(2) Kozlov, V. A.; Aleksanyan, D. V; Korobov, M. V; Avramenko, N. V; Aysin, R. R.; Maloshitskaya, O. A.; Korlyukov, A. S.; Odinets, I. L. The First Solid Phase Synthesis of Pincer Palladium Complexes. Dalt. Trans. 2011, 40 (35), 8768–8772. https://doi.org/10.1039/C1DT10680E.
(3) Gómez-Benítez, V.; Germán-Acacio, J. M.; Morales-Morales, D. Mechanochemistry a Promising Tool on the Synthesis of Organometallic Pincer Compounds. Current State and Future Perspectives. Curr. Org. Chem. 2022, 26 (5), 438–443. https://doi.org/https://doi.org/10.2174/1385272826666220214110600.
Author Response
How the authors determined that the molar ratio 4.5 (ligand): 1 (metal) was adequate for the reaction to take place. Did you explore other stoichiometries to find this relationship the right one? - The synthetic scheme is added to the text, and the proposed mechanism is discussed in the revised version. In our opinion, the used ratio (ligand): (metal) is not only sufficient, but even redundant. It is known that solid PtCl4, when heated to 380 C, decomposes with the release of chlorine. Obviously, when grinding PtCl4, especially in the presence of a reducing agent, the decomposition temperature should decrease, which is what we observe in our reaction. To interact with 1 molecule of chlorine, 1 molecule of the reducing agent is sufficient. To obtain Pt(β-diketonate)2, 2 moles of the diketone salt are needed. The ratio (ligand) : (metal) = 4.5 : 1 was used. Two moles of the ligand go to the formation of the complex, one mole goes to the reduction, and 1.5 moles were taken to ensure the completeness of the reaction. In a previous work, the ratio (ligand) : (metal) = 5 : 1 was used in the reaction with K2PtCl6 [Makhaev, V., Petrova, L. Solid-phase synthesis of platinum group metal β-diketonates. Inorg. Chim. Acta. 2021, 518, 120231. https://doi.org/10.1016/j.ica.2020.120231.
Haven't the authors used precursors with more labile ligands [PtCl2(PhCN)2] to better proceed with the reaction? - Thank you for the provided references. Note that in these studies, complete displacement of chlorine is not achieved; one or two of the chlorine atoms remains bound to the metal in the resulting complex. Moreover, the interaction with the platinum salt in these works occurs at ~ 230℃. In our case, the completeness of the chlorine displacement and the formation of final products is achieved at much lower temperatures, about 170 ℃
Have the authors explored LAG (liquid-assisted grinding) reactions? The addition of small amounts of solvent (in catalytic amounts) favors the reactions to proceed more adequately. - We know and use methods of activation in the presence of a solvent. In cases where increasing the solubility or cleaning the surface of the reagent from a poorly soluble product is the key step in the reaction, these methods are very effective. But we are developing a different technique, the solid-state mechanochemical reaction (interaction of two solid reactants). In our case, the presence of a solvent leads to dissipation of mechanical energy due to its transfer to the volatile solvent. Our approach made it possible to reveal the relationship between the degree of grinding of the reagents and the reaction rate, between purely mechanochemical reactions and reactions in the thermal explosion mode, as well as self-propagation reactions [1-4]. The main goal of this project was to develop a general method for the synthesis of beta-diketonates of platinum group metals. We solved this problem. However, we were not satisfied with a high temperature of the synthesis of platinum beta-diketonates. Therefore, the goal of this work was to find a reaction for the synthesis of platinum beta-diketonates that proceeds under milder conditions than the reactions with PtCl, or K2PtCl6 stydied earlier. To solve this problem, we studied the reactions of some sodium beta-diketonates with PtCl4. Indeed, we succeeded in significant lowering the reaction temperature in the reactions at a sufficiently high yield of products.
- A.P. Borisov, L. A. Petrova, T. P. Karpova, V. D. Makhaev. The Solid-Phase Synthesis of Chromium β-Diketonates upon Mechanical Activation. Russ. J. Inorg. Chem. (1996) V. 41. No. 3, pp. 394-399.
- V.D. Makhaev, A.P. Borisov, L.A. Petrova. Solid-state mechanochemical synthesis of ferrocene. J. of Organomet. Chem. (1999) V. 590, Issue 2, pp. 222–226. https://doi.org/10.1016/S0022-328X(99)00460-X
- V.D. Makhaev, A.P. Borisov, V.V. Aleshin, L.A. Petrova, and B.M. Zuev. Self-Propagating Synthesis of Iron(III) Acetylacetonate after Mechanical Activation of the System FeCI3-NaC5H7O2. Int. J. of Self-Propagating High-Temperature Synthesis. (2000) V. 9, No. 3, pp. 297 – 306 (and references therein).
- V.D. Makhaev, L.A. Petrova, K.A. Alferov, G.P. Belov. Mechanochemical Synthesis of Chromium Tris(2-ethylhexanoate) and Evaluation of Its Catalytic Activity in the Reaction of Ethylene Trimerization. Russ. J. Appl. Chem. (2013) V. 86, No. 12, pp. 1819−1824.
how did your determine the reaction times (3 or 4 h)? Did the authors follow up on previous reaction times to determine these reaction times. – Yes, we did. We discuss this problem in the “Results and discussion”. An increase in the grinding time often leads to the decomposition of coordination and organometallic compounds. Heating activated mixtures accelerates the reaction and increases the yield. We obtained rather good yield ca. 70%.
It would be appropriate if you mentioned the frequency of C=O of the diketonates as free ligands and once coordinated to corroborate the coordination of the ligand. – We added two Tables with the data of IR spectroscopy for sodium beta-diketonates, ground mixtures, and obtained Pt beta-diketonates to the revised version.
The brand of the ball-milling device was not specified. – We added that we used a custom-made vibration ball mill. The data are in a number of our previous articles listed in the “References”. It would be appropriate if the reviewers read not only the peer-reviewed work, but also previous works of the authors.
Reviewer 3 Report
There is a lack of clarity in the presentation of the results.
There should be a diagram describing the synthesis.
The abstract does not describe the originality or contribution of the work.
The discussion should be more profound, clear, orderly and should be compatible with the title and objective described.
The images should be more visible and enlarge the titles of the graphs.
In general the work requires severe modifications as in the description.
Observations
1) There is a lack of clarity in the presentation of the results.
In the results the authors make a description of what they observe in the FT-IR, XRD and thermal analysis, but they do not make a discussion about the results. It would be of great relevance that the authors propose a reaction diagram placing the chemical structures of the reactants and products so that they can strengthen the discussion of what is observed in the results.
2) The abstract does not describe the originality or contribution of the work.
The abstract is written in a general way, it does not refer to important results obtained such as product yields, reaction times, analysis used and the explanation why the mechanochemical method is better than other methods used for the synthesis of B dicetonates.
3) Improve the images corresponding to the thermograms as they look unclear, increase the size of the legends.
Author Response
There is a lack of clarity in the presentation of the results. In the results the authors make a description of what they observe in the FT-IR, XRD and thermal analysis, but they do not make a discussion about the results. It would be of great relevance that the authors propose a reaction diagram placing the chemical structures of the reactants and products so that they can strengthen the discussion of what is observed in the results. - The result is that the authors discovered a new synthetic reaction that greatly facilitates the preparation of platinum beta-diketonates. The synthetic scheme is added to the text, and the proposed mechanisms are discussed in the revised version.
The abstract does not describe the originality or contribution of the work. The abstract is written in a general way, it does not refer to important results obtained such as product yields, reaction times, analysis used and the explanation why the mechanochemical method is better than other methods used for the synthesis of B dicetonates. - The details are presented in the text. The main achievement of this work, a substantial lowering reaction temperature comparing to reactions with PtCl2 or K2PtCl6 is described in the abstract. Our technique makes it possible to grind the reactants for 4 h, and then sublime the product at about 170 C from dry ground mixture with a yield of ca. 70% within one day. Other methods require significantly more time, steps and reagents. The difficulties in the synthesis of platinum(II) β-diketonates are well-known and described in many works, including Inorganic Synthesis. They include a number of reagents, operations, often with corrosive medium such as hydrofluoric acid, and time. We discussed it previously. - V. Makhaev, L. Petrova. Solid-phase synthesis of platinum group metal β-diketonates. Inorg. Chim. Acta. (2021) V. 518, 120231. It would be appropriate if the reviewers read not only the peer-reviewed work, but also previous works of the authors before writing a peer-review. https://doi.org/10.1016/j.ica.2020.120231
The discussion should be more profound, clear, orderly and should be compatible with the title and objective described. - The manuscript was substantially revised in accordance with the comments of the reviewers and the requirement of the editors to submit a comprehensive manuscript with the total volume of more than 4000 words.
The images should be more visible and enlarge the titles of the graphs. - The images are corrected in accordance with the reviewer's comments.
in general the work requires severe modifications as in the description. - The manuscript was substantially revised.
Round 2
Reviewer 3 Report
I have reviewed the corrected version and agree with its publication.